Day and night camera trap videos are effective for identifying individual wild Asian elephants

Montero-De La Torre Sasha 1
http://orcid.org/0000-0002-8058-9876 Jacobson Sarah L. 1 2
Chodorow Martin 1 2
Yindee Marnoch 3 drfungy2000@yahoo.com
Plotnik Joshua M. 1 2 joshua.plotnik@gmail.com
1 Department of Psychology, Hunter College, City University of New York , New York, New York , United States
2 Department of Psychology, The Graduate Center, City University of New York , New York, New York , United States
3 Akkhraratchakumari Veterinary College and One Health Research Centre, Walailak University , Thasala, Nakhon Si Thammarat , Thailand
Vonk Jennifer
Electronic publication date: 2023 Mar 28
Publication date: 2023
Volume: 11
Electronic Location ID: e15130
Received 2022 Jul 14; Accepted 2023 Mar 6
Copyright: © 2023 Montero-De La Torre et al.
Copyright year: 2023
Copyright holder: Montero-De La Torre et al.
License: This is an open access article distributed under the terms of the Creative Commons Attribution License, which permits unrestricted use, distribution, reproduction and adaptation in any medium and for any purpose provided that it is properly attributed. For attribution, the original author(s), title, publication source (PeerJ) and either DOI or URL of the article must be cited.
License URL: https://creativecommons.org/licenses/by/4.0/

Keywords: Conservation biology, Camera trapping, Animal behavior, Asian elephants, Remote-sensing, Human-elephant conflict

Funding: US Fish and Wildlife Service Asian Elephant Conservation Fund F18AP00456; F19AP00052; F22AP00035 Research Foundation of the City University of New York Elephant Family THA-175 Golden Triangle Asian Elephant Foundation National Science Foundation Graduate Research Fellowship DGE-1646736 This work was supported by the US Fish and Wildlife Service Asian Elephant Conservation Fund (F18AP00456; F19AP00052; F22AP00035), the Research Foundation of the City University of New York, Elephant Family (THA-175), the Golden Triangle Asian Elephant Foundation, and a National Science Foundation Graduate Research Fellowship (DGE-1646736) awarded to Sarah L. Jacobson. The funders had no role in study design, data collection and analysis, decision to publish, or preparation of the manuscript.

==============================
Regular monitoring of wild animal populations through the collection of behavioral and demographic data is critical for the conservation of endangered species. Identifying individual Asian elephants (Elephas maximus), for example, can contribute to our understanding of their social dynamics and foraging behavior, as well as to human-elephant conflict mitigation strategies that account for the behavior of specific individuals involved in the conflict. Wild elephants can be distinguished using a variety of different morphological traits—e.g., variations in ear and tail morphology, body scars and tumors, and tusk presence, shape, and length—with previous studies identifying elephants via direct observation or photographs taken from vehicles. When elephants live in dense forests like in Thailand, remote sensing photography can be a productive approach to capturing anatomical and behavioral information about local elephant populations. While camera trapping has been used previously to identify elephants, here we present a detailed methodology for systematic, experimenter differentiation of individual elephants using data captured from remote sensing video camera traps. In this study, we used day and night video footage collected remotely in the Salakpra Wildlife Sanctuary in Thailand and identified 24 morphological characteristics that can be used to recognize individual elephants. A total of 34 camera traps were installed within the sanctuary as well as crop fields along its periphery, and 107 Asian elephants were identified: 72 adults, 11 sub-adults, 20 juveniles, and four infants. We predicted that camera traps would provide enough information such that classified morphological traits would aid in reliably identifying the adult individuals with a low probability of misidentification. The results indicated that there were low probabilities of misidentification between adult elephants in the population using camera traps, similar to probabilities obtained by other researchers using handheld cameras. This study suggests that the use of day and night video camera trapping can be an important tool for the long-term monitoring of wild Asian elephant behavior, especially in habitats where direct observations may be difficult.

Introduction

In the past few decades, camera trapping (using remote motion-activated cameras to collect photos and videos) has become a popular technique to study elusive and rare species with direct implications for conservation (Griffiths & van Schaik, 1993; Foster & Harmsen, 2012; Mohd-Azlan & Lading, 2006). Camera traps provide an opportunity to capture photographs and video recordings while being minimally invasive and without the need for a human operator (Griffiths & van Schaik, 1993; Swinnen et al., 2014). They can capture animal movement, activity patterns, and behaviors (Swinnen et al., 2014; Caravaggi et al., 2017; Hegglin et al., 2004; Stevens & Serfass, 2005; MacCarthy et al., 2006) that may not otherwise be observable in densely forested habitats (Griffiths & van Schaik, 1993; Foster & Harmsen, 2012). Camera trap surveys are particularly important for animal behavior researchers as they can contribute to our understanding of intraspecific social relationships and help identify environmental factors that may impact natural animal behavior (Sanderson & Trolle, 2005; Mohd-Azlan & Lading, 2006; Tobler et al., 2008; Caravaggi et al., 2017).

Camera trap technology has improved to the point where high-quality footage can be recorded to capture information about the occurrence and prevalence of species (Silveira, Jácomo & Diniz-Filho, 2003; Trolle, 2003; Hegglin et al., 2004; Stevens & Serfass, 2005; MacCarthy et al., 2006), as well as individual- and group-level activity patterns over relatively long periods of time (van Schaik & Griffiths, 1996; Gómez et al., 2005). However, in order to use cameras to study individual variation in behavior within and between animal populations, a reliable, systematic method for identifying individuals is crucial. This involves differentiating key features and characteristics that make an individual unique from other conspecifics. Many studies that identify individual animals from camera-trap photography have focused on spotted and striped carnivores with naturally-occurring markings (e.g., Karanth, 1995; Karanth & Nichols, 1998; Kelly, 2003). Looking at natural markings in animals that do not have distinct coat patterns is more labor intensive and may introduce problems concerning reliability (Goswami et al., 2012). For species without natural body markings, researchers have looked at a combination of morphological characteristics to identify individuals that include skin folds, the presence of scars, ear nicks, horn length and shape, and tail length (Laurie, 1978; Morgan-Davies, 1996).

Asian elephants (Elephas maximus) are an endangered species consisting of less than 50,000 remaining individuals on the planet (Menon & Tiwari, 2019), and are understudied in a number of scientific disciplines, including behavioral ecology and cognition (de Silva & Wittemyer, 2012; Plotnik & Jacobson, 2022). In addition, long-term studies of their ecology and behavior have been overwhelmingly limited to India (e.g., Sukumar, 1990; Vidya & Sukumar, 2005a; Srinivasaiah et al., 2019) and Sri Lanka (e.g., de Silva, Ranjeewa & Weerakoon, 2011; de Silva, Schmid & Wittemyer, 2017). This is likely due to the capacity in these countries for observing elephants by following them in field vehicles through open areas within national parks (e.g., Sukumar, 1989; de Silva, Ranjeewa & Weerakoon, 2011). Individual Asian elephants have typically been distinguished using a variety of different morphological features, such as variations in the morphology of their ears and tails, body scars and tumors, spine shape, cuts and bumps, and tusk shape and size when present (Sukumar, 1989; Goswami, Madhusudan & Karanth, 2007; Fernando et al., 2011; Goswami et al., 2012; de Silva et al., 2013; Vidya, Prasad & Ghosh, 2014). Goswami, Madhusudan & Karanth (2007) identified male Asian elephants in India using a combination of 16 different traits to reliably identify individuals. Goswami et al. (2012) later assessed different groupings of these traits to determine that “fixed morphological traits” (those which were unlikely to change over the course of a few years) were the most reliable for individual identification and in estimating the population size of male elephants. In another study, Vidya, Prasad & Ghosh (2014) used a combination of 22 traits to identify 223 individual elephants, including females. The authors demonstrated that a combination of physical traits could be used to identify individuals of both sexes and that these traits were relevant in population demographic studies requiring repeated observations of the same individuals (Vidya, Prasad & Ghosh, 2014).

These previous studies used photographs taken at multiple angles from research vehicles to identify individual elephants. While this can be highly effective when elephants are followed in open areas, it is not practical when elephants reside almost exclusively within forests (such as in Thailand and Myanmar) or when there is a need to identify elephants at night. While several studies have identified individual elephants using photographs taken from camera traps in such difficult environments (e.g., Varma, 2004; Ranjeewa et al., 2015; Smit et al., 2017; Srinivasaiah et al., 2019), these studies have not described their identification methods in detail, nor provided a measure of effectiveness for avoiding misidentification of individuals. Therefore, we provide here a detailed protocol for identifying individual elephants recorded both during the day and at night, and to determine whether stationary camera traps can provide enough information to achieve a low probability of mistakenly classifying two different elephants as the same individual.

The current study focuses on identifying individual Asian elephants within and around the Salakpra Wildlife Sanctuary in Kanchanaburi, Thailand, and on assessing the efficacy of using videos from camera traps to do so. Previous research has estimated the population in Salakpra to be between 180–200 using both camera trap photographs (Chaiyarat, Youngpoy & Prempree, 2015) and genetic analysis (Siripunkaw & Kongrit, 2005), but other estimates suggest the population may exceed 200 (Srikrachang, 2003; Chaiyarat, Youngpoy & Prempree, 2015; Department of National Parks, Wildlife & Plant Conservation, 2017). The Salakpra elephants serve not only as an important Thai breeding population (Mitchell et al., 2013), but also play an important role in the significant human-elephant conflict (HEC) occurring in western Thailand (van de Water & Matteson, 2018), during which elephants and farmers compete for access to shared habitat and crop fields. To date, no systematic identification of the elephants within the sanctuary has been conducted.

In addition to building on previous studies that have used a variety of methodologies to identify elephants from photographs, our research team’s interest in collecting behavioral data on wild elephants presented a unique opportunity to assess the utility of the video function in camera traps to identify individual animals. In the present study, we used a list of 24 physical characteristics adapted from Goswami, Madhusudan & Karanth (2007), de Silva et al. (2013), and Vidya, Prasad & Ghosh (2014) to test whether remote camera trap footage collected during the day and at night can be effective in identifying Asian elephants across a diverse landscape. We hypothesized that video camera traps would provide enough information to characterize individual elephants such that the probability of misidentification would be similar to that calculated in the aforementioned studies using hand-operated cameras during the daytime only. This identification methodology is an important step towards understanding wild Asian elephant behavior at both individual and group levels by using efficient and non-invasive camera trapping technology. If effective, the use of remote-sensing camera technology also avoids any effect researcher presence may have on the behavior (e.g., Kiffner et al., 2014) or welfare (e.g., Paul et al., 2016) of elephant study populations. The identification of individual elephants has relevance for a number of research topics, including the study of herd demography (Vidya & Sukumar, 2005b; de Silva, Ranjeewa & Weerakoon, 2011), foraging behavior (Clapham et al., 2012), and the impact of individual elephant behavior and personality on human-elephant conflict (Mumby & Plotnik, 2018; Plotnik & Jacobson, 2022).

Methods

Study area

We began studying elephant behavior in January 2019, in the Salakpra Wildlife Sanctuary, a protected area in Kanchanaburi, Thailand, in collaboration with the Thai Department of National Parks, Wildlife and Plant Conservation (DNP), the government entity responsible for managing the sanctuary and all Thai national park lands. Salakpra is approximately 868 km2 and is located within the 18,000 km2 Western Forest Complex. It is a unique protected area in that it is completely closed to tourists and permission is required to enter. The Sanctuary contains areas of mixed deciduous forests (60%), dry dipterocarp forest (30%), and disturbed land (10%) (Chaiyarat, Youngpoy & Prempree, 2015). Data in this study were collected from four different locations as part of a larger elephant behavior project: Kaeng Kaeb (KK) and Khao Seua (KS) are located within the protected area, and Tha Manao (TMN) and Mae Plasoi (MPS) are located along the periphery of the protected area near crop fields (Fig. 1). KK and KS are ranger stations within the protected area of Salakpra where, except for park ranger patrols, human activity is at a minimum. TMN and MPS are villages (specifically, crop fields along the Sanctuary’s outside border) where chances of human-elephant interactions are high. Crop fields mainly consist of corn, pumpkin, sugar cane and cassava, depending on the season.

Figure 1 Map of the study areas inside the Salakpra Wildlife Sanctuary.

The four different study sites (Kaeng Kaeb, Khao Seua, Mae Plasoi, and Tha Manao) are labeled clearly. The image on the bottom left shows the location of the sanctuary in western Thailand. The map was created using QGIS.

Permission

This study was approved by the Hunter College Institutional Animal Care and Use Committee (JP-Elephant Behavior 5/21), and permission was granted to collect data in Salakpra Wildlife Sanctuary by the National Research Council of Thailand (Plotnik 1/62) on behalf of the Thai Department of National Parks, Wildlife and Plant Conservation.

Camera trap installation

The videos analyzed in this study were recorded between February 2019 and January 2020. There were a total of 34 Browning Spec Ops Advantage remote-sensing cameras installed throughout the four sites as part of a larger behavioral monitoring project: eight in KK, 11 in KS, six in TMN, and nine in MPS (one camera from MPS was stolen in September, 2019, and was not replaced during this time period). To optimize detection and the recording of social behavior, cameras were installed around watering holes and salt licks in the protected area, and around crop fields and on pathways frequented by elephants in the villages. Camera traps were motion activated and set with a fast trigger (0.4 s) to capture 20-s high resolution video (30 frames/s) from up to ~25 m away. Videos were taken using natural light during the day and built-in infrared light at night. The cameras recorded the time, date and temperature during each recorded clip, which were automatically saved to SD cards our team collected and replaced approximately every 2 weeks.

Identifying individual elephants

In this study, there were 24 physical characteristics (Table S1) chosen to identify individual elephants and adapted from Goswami, Madhusudan & Karanth (2007), de Silva et al. (2013), and Vidya, Prasad & Ghosh (2014). These characteristics were re-defined to our specifications (see the ‘characteristics’ section below). Video clips from all four sites were first scanned and flagged for further investigation using VLC media player (version 3.0.10). For videos to be flagged, elephants had to be marked as visible and identifiable, meaning more than two characteristics were distinguishable (i.e., ear folds, tears, tail length, etc.).

Once an elephant was chosen in a flagged video, another video with the same elephant was found, primarily using videos from the same location (sanctuary or crop fields). However, in some rare instances, elephants were found to have traveled between locations (e.g., KK to KS). These videos were used to match the same characteristics, on a different date to qualify the viewed elephant as a unique individual (Fig. 2; see Montero (2020) where these procedures were originally outlined). We did not characterize an individual and add their traits to the database unless they were observed on more than one date and thus in more than one video. Elephants were also visually confirmed to be the same individual in a video from another date by looking at more specific, non-categorized details such as the shape and exact location of ear tears, shape of tail brushes, and bumps on the skin. In our study, in some instances, only one side of an elephant was visible. Therefore, when locating another instance/video, the whole view or the same side view of the elephant had to be visible in order to confidently label the images as representative of the same individual. This method was adapted from the methods used to identify individual elephants from handheld cameras in previous studies (Goswami, Madhusudan & Karanth, 2007; Goswami et al., 2019). As the purpose of the current study was to assess the efficacy of video camera trap data for identifying individual elephants, we selected footage where elephants were easily observable and thus did not need to exhaustively evaluate all of our video data.

Figure 2 Day and night snapshots of two elephants showing the variable quality of video.

(A) Shows an adult male that was distinguished by the one grown out tusk and tears on the bottom of the right ear. In the night shot of the same elephant (B), we were able to observe the top ear folds more clearly. (C) Shows an adult female with two offspring behind her, visible with her at night as well (D). This female has an especially large tear on the right ear which is distinguishable in both shots. Pigmentation on the ears and body was sometimes visible in night videos, as can be seen with this elephant.

Once an elephant was identified, it was then entered into an AirTable cloud-based database (San Francisco, CA, USA) with a unique code/number, screenshots, and associated characteristics. Characteristics were used to specify each area of an elephant’s body that could be described with different trait state options or specific features of that characteristic. For example, a characteristic such as back shape might have a trait state option such as humped to describe the characteristic. The code N/A was used when these areas of the body were not visible due to video quality or elephant body position. Video clips were matched to each elephant in the database and added continuously to record additional individual characteristics or previously unobserved trait states, as well as to monitor the elephant’s movement patterns between study areas. Male and female adult and sub-adult elephants were identified according to the age categorization outlined below. If an identified female was observed with juveniles or infants in two separate instances, the offspring were characterized and linked to the accompanying female(s) in the database.

Distinguishing individual characteristics by category

We categorized elephants into four age classes (A—adult, B—subadult, C—calves, D—infants; see Table S2 for details). All relative height differences and estimated age ranges were adapted from de Silva, Ranjeewa & Weerakoon (2011). Solitary bulls were usually coded as adults (A), as they typically leave their natal herd once mature (Sukumar, 1989; Fernando & Lande, 2000). However, in some instances, particularly when individuals we observed were among other bull elephants and height comparisons could be made, males were coded as subadult (B). In social groups, we categorized adult females by their enlarged breasts, if they were observable, or the presence of calves (de Silva, Ranjeewa & Weerakoon, 2011). Although the present study utilized age as a characteristic, the age classes mentioned are only estimates based on the trait state definitions (Table S2); we were not able to determine the exact age of individuals.

To determine the body conditions of each individual, we assessed the pelvic, shoulder, and back bones as elephants moved in a video. Body condition definitions were adapted from Fernando et al. (2009) and simplified to three categories: Zero for the underweight condition if ribs were visible, one for the normal condition if pelvic and shoulder bones were visible but ribs were not, and two for the overweight condition where pelvic and shoulder bones were not prominent (Table S3). The prominence of the backbone was also used as an indicator of body condition (Wemmer et al., 2006).

We categorized whether the individual had either tusks or tushes (incisors that are much smaller and thinner than tusks) (Kurt, Hartl & Tiedemann, 1995). In Asian elephants, only males have tusks—although not all do—while both males and females can have tushes (i.e., short tusk-like protrusions from the top of the mouth), but again, not all do (Sukumar, 1989; Kurt, Hartl & Tiedemann, 1995; Chelliah & Sukumar, 2013). When tusks were present, tusk symmetry, arrangement, and angle were recorded accordingly (Table S4). We categorized tusk symmetry based on whether the tusk length was symmetrical and tusk arrangement based on the tusk growth direction of both tusks compared to each other. We also categorized tusk angle based on the direction of the tusks in reference to a horizontal plane. We used side views of the elephant to best determine tusk angle, and the position of the trunk to help guide the decision (Fig. S1).

We described characteristics of the elephants’ ears, focusing on the left and right ear separately. Also, we considered top folds and side folds (labeled as primary and secondary fold, respectively, in de Silva et al. (2013)) as separate characteristics. We described top ear folds based on the degree to which the top ear was folded on both sides, and side folds by the way each side fold was positioned. We used the angular shape of the bottom of the ear (or ear lobes) to describe them (Fig. 3). We described other characteristics of the ears (ear tears, holes and depigmentation) when possible (Table 1). When ear tears and holes were present, we categorized their locations starting from the top to the bottom of the ear; if there were multiple tears or holes in one ear, we recorded the location with the most tears or holes. The location of other tears or holes along the ear were added as a note in the database.

Figure 3 Visual representation of some of the ear characteristics observed during the day and night.

Table 1 provides details for the different characteristics, corresponding to the labeled images in the figure (A–L). (G and J) The red circles show characteristic ear tears, and illustrate the considerable variation within this trait.

Table 1 Ear characteristics and trait state definitions.

Ear characteristics	Trait state definitions	Examples	
Ear top fold	None: when there was no true curve (fold) visible	Figure 3I	
Forward slightly: where the top of the ear was folded at an almost 90-degree angle	Figure 3C	
Forward rolling fold: where the top of the ear was folded like a ‘wave’ and we were able to still see the ear under the fold	Figures 3A, 3E and 3F	
Forward flat fold: where the top of the ear was folded so you cannot see under the fold for the majority of the ear	Figures 3B, 3G, 3H, 3J and 3K	
Backward: where the ear curved back at any angle	Figure 3L	
Ear side fold	Forward: where the side of the ear was folded forward at any angle and degree	Figures 3A, 3F, 3H, 3J and 3K	
Backward: where the side of the ear was folded backward at any angle and degree	Figures 3B–3E, 3I and 3L	
Ear lobe shape	L-angular: where the ear lobe blended in with the ear and created a wide angle	Figure 3E	
V-acute: where the ear lobe was pointed at the bottom, to form an acute angle	Figures 3C, 3D and 3H–3L	
U-rounded: where the ear lobe was more rounded than pointy	Figures 3A and 3B	
Ear tears/holes	None: no visible tear or hole seen	Figures 3A, 3C and 3D	
At. side fold: tears or holes were visible on the side folds	Figures 3J and 3K	
Before side fold: tears or holes were visible in between the top and side fold		
After side fold: tears or holes were visible between the side fold and where the bottom of the ear meets the head	Figures 3G and 3J	
On. top fold: tears or holes on the top of the ear	Figure 3J	
Ear depigmentation	Present-slight: where discoloration was seen in less than half of the ear, beginning from the bottom portion of the ear going upwards/inwards, and if little to no depigmentation was seen on the back of the ear	Figure 3L	
Present-prominent: where discoloration was seen in more than half of the ear, beginning from the bottom portion of the ear going inwards and if the majority of the back of the ear was depigmented	Figures 3B and 3C	
Note:

This table shows each of the ear characteristics coded, as well as the multiple trait state definitions for each. In addition, readers can refer to the corresponding figures listed to see the variation in the characteristics observed.

The back shape of each individual was organized into three categories (Fig. S2 and Table S5). We did not observe a ‘concave back’ in this study, but because it was described in the population studied by Vidya, Prasad & Ghosh (2014) in India, it was included as a possible category.

There were two different tail characteristics used to identify the elephants: tail length and brush type (Fig. S3 and Table S6). We categorized tail length based on the distance from the rump to the tip of the tail, not including the ‘tail brush’ or hair. We described the tail-brush type based on its length and location (i.e., the anterior side closest to the body, the posterior side farthest from the body, or both sides of the tail).

Finally, we categorized depigmentation on parts of the elephants’ bodies other than the ears if and when it occurred (Table S7). Figure S4 shows an example of an elephant with depigmentation on various parts of his body.

Interrater reliability

We assessed reliability of the categorization of elephant characteristics between the first author (SM-DLT) and another trained coder in a subset of thirty video observations, each of a different individual elephant (19 recorded during the daytime, 11 at night). Cohen’s kappas for each of 17 characteristics varied from poor to excellent between the two coders, although most characteristics had moderate or better agreement (Table S8). Although some kappas were poor, we did not exclude those characteristics in our initial analyses because ultimately the categorization used was based on multiple videos of an individual rather than the single video used in our reliability assessment. In addition, the verification of an individual’s identity relied on multiple characteristics (never just one) as well as a separate visual confirmation using other physical traits described previously. For the purposes of illustrating these points, we have included relevant analyses that include and exclude the characteristics for which there were kappas that represented poor agreement—back shape and body depigmentation—in the results below.

We further assessed the reliability of more than one coder correctly identifying the same individual in multiple videos—which aims to demonstrate the effectiveness of the ID protocol—by testing a naïve coder. This coder reviewed the written protocol but was not trained on identifying elephants, and was tasked with identifying 20.8% of the 72 adult elephants (15 individuals: eight males and seven females) from a video dataset. The coder was provided with the identification protocol including definitions of characteristics and trait states as well as each of the 15 elephants’ characteristic profiles and a reference video for each of the 15 individuals. This test mimicked the procedure that the first author followed when originally determining that a video did or did not match an elephant already identified. The coder was instructed to use the characteristic profiles to identify the 15 elephants in the video set, which included two videos of each of the 15 elephants and six videos that included only other elephants for which no other information or reference videos were provided. However, they were not told how many videos of each of the 15 elephants were in the set nor how many videos contained elephants other than those characterized. They did know that some of the videos in the set were elephants other than the 15 and were asked to identify which videos contained these “other” elephants. The first author and the additional coder agreed on the identity of the elephant, or that the video contained an elephant of undetermined identity, in 92% of the videos.

Statistical analysis

We analyzed data in Microsoft Excel (v. 2016). Using Goswami et al. (2012)’s misidentification calculation and our aforementioned system for categorizing morphological traits, we determined the likelihood of the characterization process resulting in the misidentification of two different individual Asian elephants as the same elephant by calculating the maximum probability squared (pmax2). It is important to note that for the current study, we used this calculation to determine the probability of misidentification between easily visible elephants in our subset rather than between elephants identified in all videos (Goswami et al., 2012). Also, characteristics determined from multiple video clips of each individual were used for this calculation. In our study, to determine the maximum probability that the same traits exist and can be categorized in two different elephants using the camera trap footage, we first calculated the frequency of each trait state option per characteristic and used the most frequent in calculations. For example, the most common trait state for left ear lobe shape is a v-acute ear lobe shape which was observed in 63.9% of all adult elephant sightings (Table 2). Once the most common trait frequencies were calculated, they were ranked from the most to least commonly occurring morphological characteristics with those trait states. If there was more than one characteristic and trait state option that occurred the same number of times in the populations, the first occurring characteristic as listed in the identification protocol was put first into the ranking followed by the next on the list. The characteristic list order in the protocol was arranged for capturing information from the front of an elephant’s body to the back. However, characteristics that were seen from the whole elephant like sex and body condition were placed at the beginning of this order.

Table 2 Elephant count and calculation results for pmax2 for all adult elephants (n = 72), including most to least common characteristic and trait state option.

Ranked characteristics	Majority trait state	Number of elephants with trait	Proportion	Number of elephants with combination	p max	p max 2	
Presence of tusks/tushes	None	63	0.875	63			
Back shape	Humped	63	0.875	54	0.857	0.735	
L ear hole	None	61	0.847	46	0.745	0.556	
Tail length	Below knee, above ankle	58	0.8556	35	0.652	0.425	
L ear side fold	Backward	56	0.778	29	0.618	0.381	
R ear hole	None	56	0.778	23	0.517	0.268	
R ear side fold	Backward	55	0.764	23	0.618	0.381	
R ear depigmentation	Present-Prominent	53	0.736	17	0.382	0.146	
Sex	Male	52	0.722	11	0.400	0.160	
Depigmentation on body	Both	52	0.722	10	0.348	0.121	
Body condition	1	49	0.681	8	0.320	0.102	
L ear depigmentation	Present-Prominent	48	0.66	8	0.348	0.121	
R ear lobe shape	V-acute	47	0.653	5	0.200	0.040	
L ear lobe shape	V-acute	46	0.639	5	0.348	0.121	
Brush type	Normal both	42	0.583	4	0.160	0.026	
R ear tear	At side fold	37	0.514	2	0.174	0.030	
L ear top fold	Forward rolling fold	31	0.431	2	0.160	0.026	
R ear top fold	Forward rolling fold	31	0.431	2	0.174	0.030	
L ear tear	At side fold	29	0.403	1	0.080	0.006	
Tusk symmetry	Uneven	5	0.069	0	0	0	
R tusk angle	Straight ahead	5	0.069	–	–	–	
L tusk angle	Straight ahead	4	0.054	–	–	–	
Tusk arrangement	N/A	3	0.042	–	–	–	

Exploratory statistical tests were used to determine whether characteristics were independent from each other. Independence in this case means that the traits of one characteristic cannot be predicted from the traits of another characteristic. Chi-square and Fisher exact tests were used to calculate whether the number of individuals with each combination of traits corresponded to the assumption of independence between those traits. As was the case in Goswami et al. (2012), many pairs of characteristics were not independent from one another. If traits are independent, then the probability of a combination of traits would be equal to the product of their individual probabilities. However, because of non-independence, a conditional probability calculation is more appropriate as it does not assume independence. Therefore, to estimate the probability that an individual possessed the most commonly occurring combination of traits (pmax), conditional probabilities were calculated by moving successively down the trait frequency ranking.

When computing pmax, we first calculated the probability of the most frequent trait state for presence of tusks/tushes. Next, we looked at the probability of back shape’s most frequent trait state occurring, when presence of tusks/tushes’ most frequent trait state occurred. Moving down the ranking, the next characteristic (L ear hole) and its most frequent trait state option contributed to the calculation for the probability of the L ear hole’s most frequent trait state occurring, given the presence of tusk/tushes most frequent trait and back shape’s most frequent trait. This process continued until the number of elephants with the combination of characteristics reached one (Table 2). The probability values were then squared to obtain the value for the probability of any two individuals showing the exact combination of morphological features (pmax2) (Goswami et al., 2012). pmax2 was calculated separately for all adult elephants identified, adult males identified, and adult females identified.

Results

From a total of 475 videos collected between February 2019 and January 2020, we identified 107 elephants (72 adults, 11 sub-adults, 20 juveniles, and four infants) using 24 physical characteristics and their trait state options from both day and night camera trap videos. We used the 72 adults recorded across 363 videos (56% of which were day video, 44% were night) for the calculation of pmax2 because determining elephant sex is more definitive when the elephants are sexually mature (Sukumar, 1989; Fernando & Lande, 2000). Therefore, the age class characteristic was excluded entirely, leaving 23 characteristics for the calculations. In the calculation of pmax2, the number of elephants that had the combination of traits included in the conditional probabilities decreased to zero on the 20th characteristic (Table 2). Therefore, with the inclusion of the most frequent trait for 19 characteristics, pmax2 = 0.006 for this sample (Table 2). Even when two characteristics for which there was poor interrater agreement (as discussed in the methods above—back shape and body depigmentation) were removed and thus only 17 characteristics were included, the pmax2 remained the same (0.006; Table S9).

We performed a similar calculation for pmax2 for the sample consisting of only male (N = 52), and then only female (N = 20) elephants. However, in contrast to the previous calculation for the entire sample, when calculating pmax2 for males, only 22 characteristics were included in the calculation (the characteristic of sex was excluded). After including 19 characteristics, the number of elephants with the same combination of traits reached zero (Table S10). Including 18 characteristics and their most frequent trait, pmax2 = 0.011 for the sample of male elephants. When the two characteristics for which there was poor interrater agreement were removed and thus only 16 characteristics were included, pmax2 = 0.008 (Table S11).

When performing the conditional probability calculation for females, we only included 18 characteristics, as opposed to the 22 for males, because we excluded tusk characteristics. Using the same procedure as the previous two calculations, the number of elephants decreased to zero after including 17 characteristics (Table S12). With the inclusion of 16 characteristics and their most frequent trait, pmax2 = 0.048 for the sample of female elephants. When the two characteristics for which there was poor interrater agreement were removed and thus only 14 characteristics were included, pmax2 = 0.079 (Table S13).

Discussion

The current study aimed to determine whether camera trap videos can be used to reliably identify individual Asian elephants with similar accuracy as photographs obtained from handheld, human-operated cameras. We used day and night videos to identify a total of 107 individual elephants. We calculated the probability of two individuals having the same characteristic combinations for 72 identified adult elephants within the population.

We used video footage from stationary camera traps to successfully characterize Asian elephants and our categorization provided a low probability of misidentification. A total of 19 out of 24 morphological characteristics (excluding age and four tusk characteristics) were required to reliably identify the 72 adult elephants. The pmax2 value we calculated for male elephants was similar to the value calculated by Goswami et al. (2012). When they calculated the pmax2 with ‘all traits’ included, a combination of 20 characteristics, they obtained a pmax2 = 0.010 for observed adult males. When we too only considered adult males, we found a pmax2 of 0.011 with a combination of 18 characteristics. Our pmax2 value was lower than that obtained by Goswami et al. (2012) when we considered all adult elephants (including females) using a combination of 19 characteristics (our study: pmax2 = 0.006; Goswami et al. (2012): pmax2 = 0.010).

In the present study, we also investigated the probability of misidentifying adult female elephants (pmax2 = 0.048). There may be a higher probability of misidentification for females because there were fewer females in this subset, resulting in a smaller number of trait combinations within the study population. No females possessed all of the most frequent trait states after including the 16th characteristic (Table S10). Overall, the pmax2 results for all adult elephants, only male elephants, and only female elephants illustrate that there were low probabilities of misidentification between elephants in the population, that we reliably identified the elephants in this study, and that our misidentification probabilities were similar to those calculated from photographs in a previous study conducted in India (Goswami et al., 2012). This suggests that camera trap videos captured during the day and night can provide sufficient information to characterize Asian elephants.

Importantly, this characterization has a statistically low chance of misidentification even before the final step of any identification protocol, a detailed visual comparison of elephants with the same characteristics to confirm identity. While machine learning has exciting potential to expedite the individual identification process for animal/wildlife research (Schneider et al., 2019; Vidal et al., 2021), to date, visual confirmation of phenotypic traits by one or more human observers (either in person or using remotely-captured images) remains the predominant method for researchers to identify their study subjects in the wild (e.g., Wursig & Jefferson, 1974; Bradfield, 2004; Shorrocks & Croft, 2009). Even though the misidentification calculation represents the chance of inaccurately identifying individual elephants, we emphasize that a final, visual comparison of detailed features independent of or complementary to the analyzed characteristic traits is needed. This aligns with Goswami et al. (2012)’s suggestion to incorporate visual confirmation into the identification process. This step provides further evidence of the elephant’s identity by using a number of subjectively selected traits (e.g., Moss, 1996). The current study, however, aims only to confirm that the misidentification procedure is effective when using video camera trap data to identify individual wild elephants. It is important to note that there could be a higher probability of misidentification if a population of elephants happens to share a lot of the same trait states, which could potentially happen if there is limited genetic variation within a particular population. Nonetheless, the probability of misidentification is a useful tool for evaluating whether a characteristic protocol for identifying individuals within a population is sufficient or whether more detailed morphological information may need to be added.

In our interrater reliability analysis of a subset of the data, some characteristics had low agreement between coders. In this study, elephant identification was never dependent on a single characteristic nor a single video of an individual elephant. Furthermore, characteristic categorization was never the sole determinant that the same individual was present in multiple videos; more detailed visual comparisons were also conducted. In addition, it is not surprising that some characteristics were not reliable between coders, particularly with night videos where characteristics like depigmentation and back shape can be challenging to see. We believe this supports the importance of characterizing individuals using multiple camera trap observations of the same elephant (when able to confirm visually that they are the same) and the importance of recording as many characteristics as possible.

We also recalculated pmax values with the two traits with the poorest interrater agreement (i.e., poor kappa values) removed, and still found a comparably low probability of misidentification. This supports the idea that identifying elephants reliably requires the use of multiple characteristics in combination. While we acknowledge that caution is needed when using traits in the overall identification process that may be difficult to characterize consistently across observers, their inclusion—when the probability of miscalculation is not impacted—enhances the information that can be recorded about individuals in a variety of camera trapping conditions.

We conducted an additional interrater reliability analysis with a naïve coder, and demonstrated that two observers could reliably identify the same elephant using multiple videos. This highlights the effectiveness of using a complement of a set of characteristics followed by visual confirmation using multiple videos to identify individual elephants. This methodology could serve as a model for other researchers studying different elephant populations, particularly if they are considering the use of remote camera traps.

Implications

This study supports the idea that the use of remote-sensing cameras to identify individual elephants in long-term studies of elephant behavior and ecology can be effective (Fernando et al., 2009, 2011; Goswami et al., 2012; de Silva et al., 2013; Vidya, Prasad & Ghosh, 2014) and provides a guide for identifying them using day and night videos. Given that the footage from these cameras can be as useful as photographs from research vehicles, our results hopefully can encourage researchers in other Asian elephant ranges to employ camera trap technology to systematically identify individuals in their local populations, especially in places where observing wildlife directly is challenging. As more scientists conduct research at the individual-, rather than just the population-level, this technology can better help monitor elephant movement and activity as well as characterize variation in behavior patterns between elephants (e.g., Rees, 2009; Horback et al., 2012; Sitompul, Griffin & Fuller, 2013; Horback et al., 2014).

The database of individual elephants created for the present study will be compiled into a guide and provided to Salakpra Wildlife Sanctuary rangers and local farmers as a reference for identifying resident elephants. Since frequent crop raiding has been observed around the sanctuary (van de Water & Matteson, 2018), individual identification may help provide insight into the behavior of particular elephants as it relates to their interactions with humans in the area (e.g., Cook, Henley & Parrini, 2015; Goswami et al., 2015; Ranjeewa et al., 2015; Plotnik & Jacobson, 2022). Identifying individuals from camera trap footage recorded at night may be particularly important in this context since many elephants are active in crop areas at night (e.g., Ranjeewa et al., 2015; Naha et al., 2020). A capacity for identifying elephants that frequently forage on crops could also aid in targeting HEC mitigation strategies at the individual level, a potentially more effective strategy that takes elephant behavior and personality into account (Mumby & Plotnik, 2018; Plotnik & Jacobson, 2022). More targeted strategies may help farmers manage their time as they direct their efforts towards specific individuals that they can identify within their own crop fields.

Identifying individuals is also crucially important for understanding individual variation in elephant behavior more generally. Remarkably, we know very little about such variation in wild Asian elephant behavior and how elephants adapt to rapid, human-generated environmental change (Plotnik & Jacobson, 2022). The current study helps form a foundation for future research in this area. Our own work aims to use the individual identification of wild Asian elephants to assess differences in personality and cognition, not only as a means to help in their conservation, but also as a method for understanding how flexibility in behavior facilitates adaptations to anthropogenic change (Mumby & Plotnik, 2018; Plotnik & Jacobson, 2022).

Limitations of camera traps for individual identification

The position and angle of the stationary camera traps sometimes limited our ability to collect comprehensive morphological data on the elephants. While a stationary camera provides an opportunity to capture some characteristics of an elephant in its field of view, what is captured is dependent on the elephant’s distance from the camera and movement across its view. Some videos only captured ears, backs, and tails, while other videos did not capture backs, tail length or ear top folds. Depending on the elephant’s approach and activity in front of the camera, sometimes only one side of the elephant was recorded. Camera traps deployed in the field were typically put up in a high place and were stationary for a long period of time. The only way to change the view would be to manually move the direction of the camera, and this was usually done infrequently due to their installation in remote areas. Overall, this limitation on the camera’s mobility increased the frequency of data points where the elephant could not be identified due to a lack of observable characteristics. Nonetheless, even though the position and immobility of camera traps at any given time limited the number of views of an elephant during a single observation, elephants were not included in our analysis unless we could confirm that they were the same individual in different videos based on a majority of the same traits. We suggest that future studies focusing on individual identification would benefit from installing multiple cameras together at different angles to ensure that both sides of the elephant are visible in an observation.

Collecting identification data from night videos was another challenge. During the night, when the infrared light was illuminated, characteristics like ear folds would sometimes blend in with the color of the ear, obscuring the shape and folds, making it difficult to identify the trait state. Even with some traits obscured, there were typically others visible that allowed for identification of the elephant. In the future, we would like to compare the utility of using either day or night video to identify specific traits, as well as compare how often elephants are not able to be identified depending on the lighting. This could also inform the differential installation of future cameras based on elephant movement patterns at different times of day.

While there are limitations for using camera traps to identify individual elephants, the installation of remote cameras allows for the capture of multiple screenshots from video at a much closer proximity than is usually possible when humans are present. Thus, the lack of a human presence during data collection, which could negatively impact the elephants’ behavior, may help offset the limitations camera trapping poses to individual identification. Videos also allow for determination of traits based on nuanced body movements. For example, analyzing frame-by-frame ear flapping can help identify side folds, while also making tears and holes more evident than they might be in a photographic snapshot. The use of video, when possible, could provide greater clarity than photographs during individual identification, especially when distinguishing between elephants with marked similarities in phenotype.

A limitation of identifying elephants using the characteristics discussed herein is that not all of them were temporally static. Our identification protocol included some characteristics that are temporally variable, although over different time scales (Goswami et al., 2012; Vidya, Prasad & Ghosh, 2014). For example, an individual’s body condition may change across seasons, degree of depigmentation may change across years, and new ear tears or tusk breaks could occur anytime. Researchers using this identification methodology must be aware of these potential changes and confirm the final individual identity using further analyses of the images. To maintain accurate identification, we suggest updating individual characterizations in a database as often as possible. Since the data included in the study only covered one year and no single characteristic determined a new individual, it is unlikely that any temporal changes affected the reliability of our dataset.

Conclusion

In the current study, 72 adult elephants were reliably identified through camera trap videos based on the misidentification probability calculation (pmax2) and using 19 of the 24 possible morphological characteristics (Table S1). While these characteristics were derived from previous Asian elephant identification studies (Goswami, Madhusudan & Karanth, 2007; de Silva et al., 2013; Vidya, Prasad & Ghosh, 2014), this is the first known study to evaluate the use of these characteristics for elephants in Thailand, and to do so using remote-sensing camera traps. The present study also indicates that camera trap videos are able to capture enough characteristics of Asian elephants to use this categorization protocol to identify individuals with a low chance of misidentification. Given the overall quality of these video data, future studies should focus on developing more automated and less time-intensive elephant identification processes using machine learning and other computer vision techniques (Bodesheim et al., 2022).

We hope these results will help inform the use of camera traps in the wild to study individual elephants, demographics and population dynamics and behavior. Camera traps, and video data collected from them in particular, provide a unique opportunity to record animal behavior over a cumulatively long period of time without the negative impacts posed by human presence or interference while filming. This is particularly important for the relatively new study of conservation behavior (the use of animal behavior research in conservation practice) and the application of animal behavior research to human-wildlife conflict. For elephants in particular, understanding individual differences in elephant behavior and how elephants make decisions about risk may have important implications for mitigating human-elephant conflict (Plotnik & Jacobson, 2022). The elephants’ decision-making process and differences in how they behave in and around human-dominated landscapes can best be observed from a viewpoint that minimizes the impact of researcher presence or behavior on the elephants. We believe that remote-sensing camera traps present a unique and exciting avenue for collecting such data, and encourage scientists interested in wildlife behavior and its application to conservation to consider the use of remote video-recording devices in their own work.

Supplemental Information

Supplemental Information 1 Visual representation of some of the different tusk arrangements observed.

Table S4 provides detailed definitions/descriptions that correspond with each of the labeled images (Figs. S1A–S1E). The quality of the images is reduced due to their capture from video.

Click here for additional data file.

Supplemental Information 2 Visual representation of prominent back shapes.

See Table S5 for descriptions of the back characteristics corresponding to the labeled images in the figure (Figs. S2A–S2C).

Click here for additional data file.

Supplemental Information 3 Visual representations of tail length and brush type.

See Table S6 for descriptions of tail characteristics corresponding to the labeled images in the figure (Figs. S3A–S3H).

Click here for additional data file.

Supplemental Information 4 A bull elephant with depigmentation.

Note the different shades of pink on the elephant’s ear and underside of the trunk.

Click here for additional data file.

Supplemental Information 5 Supplementary Tables.

Click here for additional data file.

Supplemental Information 6 Characteristics of each identified elephant (raw data).

The raw data coded, using specific characteristics, for each individual elephant identified. Note that most characteristics are not coded for elephants in the C & D age categories unless they had distinct characteristics, however, these age groups were not included in our pmax analyses.

Click here for additional data file.

We thank two anonymous reviewers, Jennifer Vonk, and Emily Polla for their helpful feedback on this manuscript. We sincerely thank Parntep Ratanakorn for his facilitation of our collaborations during this study. We also thank the National Research Council of Thailand, the Thai Department of National Parks, Wildlife and Plant Conservation, and the Salakpra Wildlife Sanctuary (particularly the sanctuary chief, Paitoon Intarabut, and his staff) for allowing us to conduct this research. We are appreciative of Alexander Godfrey’s assistance with mapping and camera trap placement. Thank you to our field team in Thailand (Weerach Charerntantanakul, Pornpimol Kubsanit, Juthapathra Dechanupong, Wantida Horpiencharoen, and Prawit Innoy) for collecting all the videos from the field and maintaining the camera traps, and to the park rangers and local community members in Kanchanaburi, Thailand that provided invaluable assistance and support during this study. Thank you to Varun Goswami for guidance on the analyses performed in this study, and to Scott Gulizio and Sabana Gonzalez for their assistance with coding for interrater reliability. This manuscript was adapted, in part, from SM-DLT’s Master’s thesis in the Animal Behavior and Conservation Program in the Department of Psychology at Hunter College, City University of New York (Montero, 2020), and an earlier version of this manuscript was preprinted (Montero-De La Torre et al., 2021: https://ecoevorxiv.org/wj8p7).

Additional Information and Declarations

Competing Interests

Author Contributions

Animal Ethics

Field Study Permissions

Data Availability

Joshua M. Plotnik is the founder of Think Elephants International, Inc.

Sasha Montero-De La Torre conceived and designed the experiments, performed the experiments, analyzed the data, prepared figures and/or tables, authored or reviewed drafts of the article, and approved the final draft.

Sarah L. Jacobson conceived and designed the experiments, performed the experiments, analyzed the data, prepared figures and/or tables, authored or reviewed drafts of the article, and approved the final draft.

Martin Chodorow analyzed the data, authored or reviewed drafts of the article, and approved the final draft.

Marnoch Yindee conceived and designed the experiments, authored or reviewed drafts of the article, and approved the final draft.

Joshua M. Plotnik conceived and designed the experiments, performed the experiments, prepared figures and/or tables, authored or reviewed drafts of the article, and approved the final draft.

The following information was supplied relating to ethical approvals (i.e., approving body and any reference numbers):

This study was approved by the Hunter College Institutional Animal Care and Use Committee (JP-Elephant Behavior 5/21), and permission was granted to collect data in Salakpra Wildlife Sanctuary by the National Research Council of Thailand and the Thai Department of National Parks, Wildlife and Plant Conservation.

The following information was supplied relating to field study approvals (i.e., approving body and any reference numbers):

The National Research Council of Thailand.

The following information was supplied regarding data availability:

The raw data is available in the Supplemental File.

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
