# Peer review of "Day and night camera trap videos are effective for identifying individual wild Asian elephants"

_PeerJ, doi:10.7717/peerj.15130_

## Round 0.1 · original submission · Major Revisions

I was fortunate to receive three very helpful reviews of your manuscript. Two of the reviewers point to the need to clarify how your study adds to or builds upon existing research that has already validated the use of video for identifying elephants. Just as critically, the reviewers point to the importance of inter-observer reliability. A revision of the manuscript cannot be considered for publication unless you are able to address these two major concerns. The reviewers also point to the need to address issues of the representativeness of the sample and the images. I agree with them that it will be important to understand how often elephants could NOT be recognized from the images, and how changes in body quality, etc. can be addressed. If you can address these, and the other issues identified by the reviewers, the revision can be considered.

Reviewer 1 ·

Basic reporting

I enjoyed reading this well-written and -prepared article. I do have some suggestions for how I think it may be improved to benefit readers and other researchers interested in this topic.

A) Lines 117-120: You note that other studies have identified individual elephants in forest environments, but you then state that more research on this topic is needed. I, and I believe other readers, would appreciate more clarity here on why it is that you believe more research on this topic is needed. Could you perhaps briefly summarize what flaws or limitations exist in the prior research that your work is intended to address?

B) Please ensure that all references cited in the text are present in the list of references, and vice versa. For example, Griffiths & van Schaik (1993a) is cited on lines 56 and 62, and Mohd-Azalan & Lading (2006) is cited on lines 56-57 and 65, but I cannot find those articles in the References. I have not reviewed the rest of the articles cited in the text to ensure that they are in the list of references, nor have I verified that all articles listed in the References are cited in the text. Please do so.

C) Please review the list of references for consistency. Most references appear consistent, but perhaps not all. For example, in the peer-reviewed reference MacCarthy et al. (2006) and several others, but not all others, all nouns are capitalized.

D) You may need to verify image quality before publication. For example, Figure 1 appears to be 200dpi, not 300dpi.

E) Figure 1: The study areas are abbreviated in the Legend, but not on the map. To assist readers, I therefore suggest either including both the study area names and abbreviations in the Legend, or else label the map with just the study area abbreviations.

Experimental design

A) You do a thorough job of illustrating how the combination of traits allows you to identify individual elephants in your sample set. However, I believe that this assumes that each observer would gather the same information from each video. I would appreciate some evidence that if different observers independently reviewed the same videos, they would independently assign the same traits to the elephant in those videos. One way in which you might demonstrate this would be to have multiple observers blindly score the traits demonstrated in a set of videos (perhaps n=30), then calculate the interobserver reliability for each of the traits (e.g., Viera, A.J. & Garrett, J.M. 2005. Understanding interobserver agreement: the kappa statistic. Fam. Med. 37, 360–363.) You may learn that there is less interobserver agreement for some traits, which could lead them to be less reliable for individual identification.

B I think it’s important that you clarify some of your methods. In lines 190-194 you state that you identifed individuals in videos, then found other videos showing the same individuals. How did you know that two videos showed the same individual? Similarly, on lines 414-419 you note that you used “a majority of the same traits” to confirm whether two videos showed the same individual. Could you clarify this? If I understand correctly, the logic appears circular: you used similarity of appearance to identify the same individual in multiple videos, then you used multiple videos of the same individual to identify combinations of traits unique to that individual.

C) On lines 226-232 you describe how body condition was evaluated, as one of the characteristics used to identify individuals. Might not body condition change over time due to various factors (e.g., season, disease, age), making it unreliable as a trait used to identify individuals?

D) In lines 404-419, you describe the challenges posed by using a fixed orientation camera, which may often not record all useful identification information. Could you improve the probability of obtaining more information from each visit by an elephant to sampling location if more than one camera was located at each sampling location? If so, I think this would be worth suggesting in the Discussion.

Validity of the findings

No comment.

Additional comments

A) Were you surprised that the sex ratio of identified adult M:F was 52:20? Why do you think there was such a biased sex ratio in your sample set? Could the factors behind that bias influence your results?

B) I suggest that you consider shortening your title by deleting “Day and night”.

C) Introduction, lines 104-111: Although individual identification is a key element of capture-recapture methods, which can be very productive research methods, I don’t think you need to provide quite so much information on capture-recapture methods.

D) Line 156: I suggest deleting “identified as”.

E) Lines 159-160: I suggest that you move the sentence “The Sanctuary…” to line 152, before the sentence “Data in this study…”

F) Line 172: I suggest deleting “In order”.

G) Lines 338-340: I suggest deleting the sentence “Table 2, Table S8…”, because it seems redundant with the information presented on lines 320-337.

H) Line 490: I suggest deleting “would like to”.

I) Line 490: I suggest deleting “would” and “like to”.

J) Line 639: Please capitalize “one”.

Reviewer 2 ·

Basic reporting

The article is well written in terms of the language and coherence, and the overall writing style. Necessary literature are also appropriately cited, but, where the article fails is in having a hypothesis, and how or whether it was tested.

Experimental design

1. The overall research question of the study is a bit unclear, except that the authors indicate that they wanted to assess efficacy of video cameras in recording individual elephants. There is no research gap here that is being filled by the study.
2. There are earlier studies that have extensively used camera traps in recording elephant photographs, identifying individual elephants, and even assessing their by examining association patterns through photographs and videos (authors have cited one of those studies; Srinivasaiah et al. 2019). Hence, assessing efficacy of the method seems irrelevant, especially since it has been well established earlier.
3. In methods section, it is unclear whether same observer coded all the data and identified all the individuals, considering there could be inter-observer variability in validating individual ID.
4. Most importantly, when an individual elephant is being identified and recorded, both profiles, right and left, are to be observed, as in the case of even carnivores (explaining the importance of paired camera units, placed diagonally opposite to each other). And it is not too often that video cameras record both flanks. Hence, identification, unless recorded from both sides remains incomplete, and could also possibly lead to misidentification of individuals with similar features.

Validity of the findings

It is unclear at this point what the authors set out to find in this study. Before any replication, there needs to be clarity on how videos are different/superior from hi-resolution photographs recorded by camera traps, especially when two units placed at two different angles could possibly record both profiles of an animal and provide more robust information.

Additional comments

It seems like the authors had a subset of data from a larger dataset that they wanted to analyse separately and make into a manuscript. However, the data presented here lacks a hypothesis, does not clearly state what contribution it makes to science (in addition to all the earlier studies that have used camera traps to study and record individual elephants and their behaviour), and does not clearly state what advancement it makes in the field of (elephant) population ecology. The methodology itself needs to be clearly explained, as to why multiple profiles of all individuals were not taken into consideration, and whether inter-observer variability was accounted for while identifying individuals from prerecorded videos. In the light of this, the article in its present state may not be a significant contribution to the advancement of the field.

·

Basic reporting

This article is comprehensively written, with well structured sentences used throughout and avoidance of unnecessary jargon. Literature is referenced appropriately and adequately covers the field, including referencing in the introduction supporting conservation benefits for the usefulness of developing this individual identification method for Asian elephants. Supplementary materials and raw data are provided in an easily comprehendible format.

Experimental design

Authors clearly outline the aims and the original research element of the study.

The experimental design is soundly structured using referenced literature as a basis for the physical characteristics and methodology/statistical analysis used. Methods are described clearly and with sufficient detail to successfully replicate the study.

Good definitions for characteristics are provided (e.g. Table S1), with further explanations provided in supplementary tables where required.

Validity of the findings

The statistical analysis and results sections are clearly written and easy to understand. The statistical analysis seems appropriate for the data. Basis for method previously established in published manuscript. Limitations of sample size, especially in relation to female elephants, was well captured in the discussion.

Was consideration given to how to accurately identify individuals using characteristics such as body condition that may change over time for long-term studies? Further discussion regarding this may be beneficial. See Additional Comments section for relevant line numbers.

Regarding the practical applications of trail cameras for readers considering using this method in future studies, more information regarding how often elephants could not be identified using the camera trap method may be beneficial. See Additional Comments section for relevant line numbers.

Additional comments

Suggested changes:

Abstract - given the interest in research with machine learning/artificial intelligence in image detection/footage review, it could be beneficial to make clear in the abstract that the camera traps were used by humans to identify individual elephants, as you have mentioned on line 274.

140 -141 - you could expand on the benefits of the importance of using non-invasive technology by referencing literature commenting on the welfare and behavioural implications of using more invasive methods.

228-234 – was consideration given to how to accurately ID individuals using traits such as body condition that may change over time for long-term studies? Further discussion regarding this may be beneficial.

242 – is the potential for tusk snapping to be accommodated in the tusk length metric? If so, please describe or mention how this variable can change suddenly.

252-256 – is there any literature available regarding how pigmentation and tares on ears are likely to change over time? E.g. in lions, the pigmentation spots on their noses increase over time until the whole surface is almost black, making it an unreliable individual identifier over time. Any literature available to support the reliability of ear tears or pigmentation over time in this species would be useful here.

262-270 – useful to note if these metrics would be expected to change over time and how that change may be accounted for in the design of your model.

276-279 – how do the easily visible elephants used for this calculation compare to the visibility of animals in the majority of gathered data? Does this calculation accurately reflect how this method might be used in the field?

416-417 – is there a value you could report on this to give readers a more precise understanding of how often elephants could not be identified using the camera trap method? This information may benefit researchers considering the efficacy of different methods of capturing individual identification information.

---

## Round 0.2 · Minor Revisions

Thank you for the excellent work you have done to clarify the contribution of the article. I continue to find the paper well written and believe it could be impactful for research in this area. However, it is still a bit confusing how the reported work provides evidence that individual elephants can be reliably identified using this technique. It seems that the findings may not generalize to other populations if the variability of traits differ in other elephant populations. If I understand correctly, you have shown that traits can be identified (although reliability is poor for some traits) and that these traits are unlikely to predict the identity of more than one elephant in the study group. But I am missing whether the identification of these traits reliably allows multiple observers/raters to predict the same unique individual. Apologies if that information is in the manuscript. Perhaps you just need to make this additional finding more explicit. Both of the reviewers indicate major revisions are needed to clarify some additional points about inter-observer reliability but I don't think you need to rewrite so much as to add some additional information, which is why I have decided on minor revisions.

Reviewer 1 ·

Basic reporting

In general, the basic reporting is good, although I have a few small suggestions:

Line 126: Please revise “studies to identify” to something like “studies identified”.

Lines 357-364 appear to belong in the Results section, not in the Methods section.

Table S2: I suggest changing the trait state definition of Adults from “had enlarged breasts if female or presence of calves” to “if female had enlarged breasts or calves present”

Experimental design

You’ve done a better job in the Introduction of describing why this work is needed and clarifying the purpose of this manuscript; thank you.

I have a question about the characteristic of ear tears/holes. Trait states seem to be mutually exclusive, expect for ear tears/holes. Were the trait states for ear tears/holes mutually exclusive, or could the same ear fit multiple categories? For example, could the same ear have holes in the side folds (“at side fold”) and between the top and side fold (“before side fold”)? If so, I think this would be worth considering in the manuscript.

Lines 261-263: Given that no data, and no interobserver reliability scores, are available for the “more specific, non-categorized details such as the shape and exact location of ear tears, shape of tail brushes, and bumps on the skin”, I believe it will be a challenge for readers to be confident in the use of those details for confirmation of ID. Perhaps the utility of identifying an elephant in one video (line 253-255, line 525), and then in another video (lines 256-261), is just not clear enough to me in the text, at present. For example, because this manuscript is focused on methods, what could you conclude, methodologically, from the identification of an individual in two videos that you could not conclude from the identification of an individual in one video that you never identified in a second video? If you recorded a unique set of trait states in one video, but never again, would you identify that video as one detection of a unique individual, or would you ignore the combination of trait states recorded in that single video?

Validity of the findings

I understand that values of kappa were generated from only 1 video/elephant used in rater reliability assessment, and that review of multiple videos may provide more reliability for that trait (lines 359-364, 518-522), by filling in the gaps or confirming character trait sets for individuals. However, this seems to assume that repeated recordings of the same individual can be reliably grouped together (e.g., lines 373-374), which I believe you have not clearly described or demonstrated to be the case. This is particularly so considering that your overall analyses depend on scoring footage in which elephants were easily observable (lines 275-276). If your resampling of some of the highest quality videos shows low kappa values for several traits, how can you be confident that these traits can be reliably scored in videos of lower quality? For example, I see that back shape was the second characteristic that entered into calculating Pmax for all elephants (Table 2), yet Cohen’s kappa for back shape is 0.15 (Table S8, “poor”). If you dropped from analyses the traits with fair or poor interobserver reliability, would you still achieve a sufficiently low probability of misidentification?

Additional comments

I appreciate the work you’ve done to address my prior comments. However, some aspects of the Methods are still unclear to me (i.e., how to reliably group videos from the same individual as suggested on lines 261-263) and I’m concerned about the impact on individual identification of traits with low interobserver reliability.

·

Basic reporting

No additional comments.

Experimental design

Lines 518 - 526: The inter-rater reliability assessment used single videos of each individual whilst the study used multiple videos of an individual. The inter-rater reliability method would benefit from being reflective of the study methodology so you are testing the reliability of what was measured.

Should the results of the inter-rater reliability be referred to in the results section, rather than the methods?

Validity of the findings

Lines 616 – 626 comments on the limitations of the ID method used in this study which is beneficial, however, more could be done to suggest whether this method is reliable. Comparison to data for elephants in the study region (if available) showing the frequency with which new ear tears or tusk breaks occur would give an indication of how much influence the changes in these variables are likely to have regarding the accuracy of this ID method. This should also be considered for body condition and ear/body depigmentation, particularly given that these variables had fair to poor levels of agreement during interrater reliability testing.

Additional comments

No additional comments.

---

## Round 0.3 · accepted · Accept

Thank you for undertaking the additional work to demonstrate the reliability of these methods. I have just a couple of very minor comments that I think you can address at the proofing stage.

Please ensure you are consistent with the ordering of references within citations. I found one set of references that is neither alphabetically or chronologically ordered.

Lastly, please replace any & in the main text with "and". Use & only within parentheses.

I look forward to your article being published.